The DBCV index is more informative than DCSI, CDbw, and VIASCKDE indices for unsupervised clustering internal assessment of concave-shaped and density-based clusters

http://orcid.org/0000-0001-9655-7142 Chicco Davide 1 2 davide.chicco@gmail.com
Sabino Giuseppe 1
Oneto Luca 3
http://orcid.org/0000-0002-2705-5728 Jurman Giuseppe 4 5
1 Dipartimento di Informatica Sistemistica e Comunicazione, Università di Milano-Bicocca , Milan , Italy
2 Institute of Health Policy Management and Evaluation, University of Toronto , Toronto, Ontario , Canada
3 Dipartimento di Informatica Bioingegneria Robotica e Ingegneria dei Sistemi, Università di Genova , Genoa , Italy
4 Department of Biomedical Sciences, Humanitas University , Milan , Italy
5 Data Science for Health Unit, Fondazione Bruno Kessler , Trento , Italy
Wan Shibiao
Electronic publication date: 2025 Aug 29
Publication date: 2025
Volume: 11
Electronic Location ID: e3095
Received 2025 Mar 26; Accepted 2025 Jul 9
Copyright: © 2025 Chicco et al.
Copyright year: 2025
Copyright holder: Chicco et al.
License: This is an open access article distributed under the terms of the Creative Commons Attribution License, which permits unrestricted use, distribution, reproduction and adaptation in any medium and for any purpose provided that it is properly attributed. For attribution, the original author(s), title, publication source (PeerJ Computer Science) and either DOI or URL of the article must be cited.
License URL: https://creativecommons.org/licenses/by/4.0/

Keywords: Density-based clustering validation index, Clustering, Unsupervised machine learning, Machine learning, DBSCAN, Internal clustering assessment, DBCV, Cluster analysis

Funding: Italian Ministero Italiano delle Imprese e del Made in Italy, Digital Intervention in Psychiatric and Psychologist Services (DIPPS) F/310240/01-04/X56 “Innovation Agreements” (Accordi per l’Innovazione) Ministero dell’Università e della Ricerca of Italy, “Dipartimenti di Eccellenza 2023-2027” ReGAInS grant assigned to Dipartimento di Informatica Sistemistica e Comunicazione at Università di Milano-Bicocca Davide Chicco’s work is funded by the Italian Ministero Italiano delle Imprese e del Made in Italy under the Digital Intervention in Psychiatric and Psychologist Services (DIPPS) (project code F/310240/01-04/X56) programme within the framework “Innovation Agreements” (Accordi per l’Innovazione) and is supported by Ministero dell’Università e della Ricerca of Italy under the “Dipartimenti di Eccellenza 2023-2027” ReGAInS grant assigned to Dipartimento di Informatica Sistemistica e Comunicazione at Università di Milano-Bicocca. There was no additional external funding received for this study. The funders had no role in study design, data collection and analysis, decision to publish, or preparation of the manuscript.

==============================
Clustering methods are unsupervised machine learning techniques that aggregate data points into specific groups, called clusters, according to specific criteria defined by the clustering algorithm employed. Since clustering methods are unsupervised, no ground truth or gold standard information is available to assess its results, making it challenging to know the results obtained are good or not. In this context, several clustering internal rates are available, like Silhouette coefficient, Calinski-Harabasz index, Davies-Bouldin, Dunn index, Gap statistic, and Shannon entropy, just to mention a few. Even if popular, these clustering internal scores work well only when used to assess convex-shaped and well-separated clusters, but they fail when utilized to evaluate concave-shaped and nested clusters. In these concave-shaped and density-based cases, other coefficients can be informative: Density-Based Clustering Validation Index (DBCVI), Compose Density between and within clusters Index (CDbw), Density Cluster Separability Index (DCSI), Validity Index for Arbitrary-Shaped Clusters based on the kernel density estimation (VIASCKDE). In this study, we describe the DBCV index precisely, and compare its outcomes with the outcomes obtained by CDbw, DCSI, and VIASCKDE on several artificial datasets and on real-world medical datasets derived from electronic health records, produced by density-based clustering methods such as density-based spatial clustering of applications with noise (DBSCAN). To do so, we propose an innovative approach based on clustering result worsening or improving, rather than focusing on searching the “right” number of clusters like many studies do. Moreover, we also recommend open software packages in R and Python for its usage. Our results demonstrate the higher reliability of the DBCV index over CDbw, DCSI, and VIASCKDE when assessing concave-shaped, nested, clustering results.

Introduction

Clustering internal evaluation. Evaluating clustering outcomes is one of the primary challenges in clustering analysis. The primary goal of clustering is to uncover hidden structures within data, enabling better understanding and decision-making. However, the absence of ground truth labels makes evaluating clustering results a challenging task. Validation of clustering can be categorized into three types: external, internal, and relative methods (Jain & Dubes, 1988). External validation methods, such as the Adjusted Rand Index, assess the clustering results by comparing them to a known clustering solution. However, since clustering is inherently an unsupervised method without any predefined ground truth, external measures are often impractical. In practical applications, internal and relative validation measures are more commonly used. Internal validation metrics assess the clustering quality based solely on the data at hand. These metrics analyze properties such as cluster compactness, separation, and density to determine how well the algorithm has grouped similar data points while keeping dissimilar points separate. Relative validation metrics, a subset of internal metrics, allow for the comparison of different clustering solutions to identify the best one. In the context of density-based clustering, clusters are viewed as regions with high data density separated by areas of low density. Despite the extensive research on relative validation measures, little attention has been given to evaluating density-based clustering results.

Clustering internal metrics. Several metrics for internal clustering evaluation exist: the Silhouette coefficient (Rousseeuw, 1987), Davies-Bouldin index (DBI) (Davies & Bouldin, 1979), Calinski-Harabasz index (CHI) (Calinski & Harabasz, 1974), Dunn index (Dunn, 1974), Shannon entropy (Shannon, 1948), and Gap statistic (Tibshirani, Walther & Hastie, 2001) are the most commonly employed statistics in biomedical informatics clustering studies, to the best of our knowledge. Even if popular, these scores are capable of assessing only convex-shaped clusters, and can be misleading when used to assess concave-shaped, nested clusters instead (Monshizadeh et al., 2022). For this reason, here in this study we focus on the Density-Based Clustering Validation (DBCV) index, and compare its results with the results obtained by other density-based clustering metrics such as the Density Cluster Separability Index (DCSI) (Gauss, Scheipl & Herrmann, 2023), Compose Density between and within clusters (CDbw) (Halkidi & Vazirgiannis, 2008; Halkidi & Vazirgiannis, 2002), and the Validity Index for Arbitrary-Shaped Clusters based on the kernel density estimation (VIASCKDE) (Senol, 2022).

DBCV (Davoud et al., 2014) was introduced in 2014 for solving this problem and correctly assessing the concave-shaped and nested clusters. The DBCV index evaluates how well the identified clusters reflect the underlying data structure by measuring the density of points within clusters, both locally and globally. This approach helps determine if the formed clusters are meaningful and accurately represent the true structure of the data, rather than being artifacts of the clustering algorithm. By leveraging the DBCV index, researchers and practitioners can gain deeper insights into the effectiveness of their density-based clustering models and ensure robust and reliable clustering outcomes.

Literature review. To provide a comprehensive overview of the current advancements in density-based clustering evaluation, this section reviews recent studies and methodologies. The focus is on the application of the DBCV metric, a pivotal tool for assessing the quality of clustering results. The DBCV metric’s ability to measure both local and global point density within clusters ensures a more accurate reflection of the underlying data structure. This section highlights the utility and superiority of DBCV in various domains, illustrating its effectiveness in improving clustering outcomes and addressing the limitations of traditional validation metrics.

DBCV has been employed in several biomedical informatics studies in the past. Recent studies have identified polygenic scores (PGS) clusters associated with specific genetic characteristics and treatment patterns, demonstrating the effectiveness of advanced clustering approaches such as Uniform Manifold Approximation and Projection (UMAP) and density-based spatial clustering of applications with noise (DBSCAN) with the DBCV index used as the validation metric (Lu et al., 2023). Beyza et al. (2021) developed a privacy-preserving density-based clustering protocol using DBSCAN with secure two-party computation. There, the quality of clustering is evaluated with measures such as the Adjusted Rand Index, the Silhouette coefficient, and the DBCV, the latter proving superior for evaluating density-based clustering because it also considers noise, providing a more accurate assessment compared to traditional metrics.

Density-based clustering cab be utilized to identify regions of high density separated by regions of low density (Chowdhury & de Amorim, 2019). A new density-based clustering algorithm employs the concept of reverse nearest neighbor and a single parameter, which can be estimated using a clustering validity index. DBCV has proven superior for assessing the quality of density-based clustering, surpassing traditional indices like the Silhouette Width and Dunn’s Index, as it measures the relative density connection between entities (Chowdhury & de Amorim, 2019). Machine learning techniques are employed to automatically identify and evaluate subtypes of hospitalized patients using routinely collected data, such as the National Early Warning Score 2. An iterative hierarchical clustering process, including dimensionality reduction via UMAP and clustering with HDBSCAN, uses the DBCV metric to evaluate clustering quality, showing superior performance compared to other validation metrics (Werner et al., 2023).

Clustering algorithms can also be employed for white matter tract segmentation, analyzing various approaches including voxel-based, streamline-based clustering, and atlas-based methods (Joshi et al., 2024). Among the evaluation metrics, the DBCV emerged as a key measure, providing an accurate assessment of the quality of density-based clustering (Joshi et al., 2024).

Unlike other validation metrics, DBCV accounts for both the internal density of clusters and the density between clusters, making it particularly suitable for algorithms like DBSCAN. Its ability to consider noise and density variability in the data makes it superior to metrics such as the Silhouette index and the Davies-Bouldin index (Kumar & Aggarwal, 2019). A study utilized quality metrics and ensemble strategies to enhance optimization and reduce computational load. The effectiveness of the DBCV index was demonstrated, showing promising performance regardless of cluster shape and in managing noisy datasets. The approach leverages the integration of multiple partitioning solutions to achieve more robust and effective results (Zhu, Xu & Goodman, 2020).

Process mining is an emerging discipline aimed at extracting process-based knowledge from event logs collected by business systems. Literature analysis using text mining and machine learning techniques has identified and predicted trends in key research areas. Using BERTopic and validating the results with the DBCV score, 49 topics were derived with a DBCV score of 0.366. This methodology demonstrated that DBCV is an effective metric for validating density-based clustering, ensuring accurate and non-overlapping cluster representation (Park, Cho & Lee, 2024).

An approach combining machine learning and psychology offers new perspectives on the conceptualization of psychological processes. Utilizing UMAP and Hierarchical Density-Based Spatial Clustering of Applications with Noise (HDBSCAN), evaluation is based on the number of clusters produced and the unclustered points. The decision to forego application-agnostic measures like DBCV, in favor of criteria closely tied to the research question, yielded more relevant results (Herderich, Freudenthaler & Garcia, 2024). For extracting topics related to COVID-19, UMAP was applied to reduce the dimensionality of vector representations of hashtags, followed by clustering with HDBSCAN. A grid search identified the model with the highest relative validity score, using a fast approximation of DBCV to evaluate density-based and arbitrarily shaped clusters. The resulting clusters represent COVID-19-related topics, with topic vectors defined as the weighted mean of the hashtag vectors within the cluster (Liu, 2022). DBCV and DBSCAN were recently employed to identify significant clusters of patients with neuroblastoma from small datasets, too (Chicco, Oneto & Cangelosi, 2025).

This study. The original short article on the DBCV index (Davoud et al., 2014) compares the DBCV index only with Silhouette Width Criterion (SWC), Variance Ratio Criterion (VRC), Dunn index, Maulik-Bandyopadhyay index (MB), and CDbw, and does so by only employing synthetic invented data. Here instead, we demonstrate its superiority also to DCSI (Gauss, Scheipl & Herrmann, 2023), and VIASCKDE (Senol, 2022), and confirm its higher efficacy with respect to CDbw (Halkidi & Vazirgiannis, 2008). We do so by utilizing not only artificial data but also real-world medical data from electronic health records. In this project, we explain why the DBCV index is more informative than other six common metrics for internally evaluating results produced by clustering algorithms, when the clusters are concave-shaped or nested.

After this introduction, we introduce and explain the mathematical properties of the DBCV index, we list DBCV software packages in R and Python, and we explained the techniques we used to validate the results of DBCV in ‘Methods’. We then report and describe the results obtained through our approaches on artificial data and real-world medical data in ‘Results’. Finally, we report a discussion of the results obtained and outline some conclusions in ‘Discussion and Conclusions’.

Methods

In this section, we first explain the original formula of the DBCV index in ‘The DBCV Index’ and we list the main R and Python packages in ‘Software packages’. We then briefly describe the density-based clustering methods that we employed in this study (‘Clustering methods’) and the the validation approaches we utilized: the validation procedure on artificial data (‘Validation on artificial data clustering trends’) and the validation phase on real-world medical data (‘Validation on real-world medical data clustering trends’).

Information regarding mathematical details of the other coefficients employed here can be found in original studies on these scores: DCSI (Gauss, Scheipl & Herrmann, 2023), CDbw (Halkidi & Vazirgiannis, 2008), and VIASCKDE (Senol, 2022). The DBCV and VIASCKDE scores range within [−1;+1], while DCSI has a range of [0,1], and CDbw has values in the [0,+∞) interval. Among the four scores studied here, CDbw is the only coefficient which does not have an upper limit. For all these four scores, higher values indicate better clustering quality (Table 1).

Table 1 Recap of the analyzed metrics for internal clustering assessment.

Metric	Original article	Interval	Meaning	Software package	
CDbw	Halkidi & Vazirgiannis (2008)	[0, +∞)	The higher, the better	Lashkov (2019)	
DBCV	Davoud et al. (2014)	[−1,+1]	The higher, the better	Siqueira (2023)	
DCSI	Gauss, Scheipl & Herrmann (2023)	[0, 1]	The higher, the better	Gauss (2023)	
VIASCKDE	Senol (2022)	[−1,+1]	The higher, the better	Senol (2024)	
Note:

CDbw, Compose density between and within clusters; DBCV, density-based clustering validation; DCSI, density cluster separability index; VIASCKDE, validity index for arbitrary-shaped clusters based on the kernel density estimation.

The DBCV index

The DBCV is an index designed to internally evaluate the quality of density-based clustering algorithms (for example, DBSCAN (Ester et al., 1996) and HDBSCAN (McInnes, Healy & Astels, 2017)). It extends the concept of the Silhouette coefficient (Rousseeuw, 1987) by replacing the usual distance (that is, Euclidean) with density-based distances (or connectivity measures). Let X={x1,x2,…,xn} be a dataset of n objects in Rd, and let C={C1,…,Ck} be a clustering result with l clusters.

The all-points-core-distance for an object x in cluster Ci is defined as:

APCD(x)=(1|Ci|−1∑j=2|Ci|(1KNN(x,j))d)−1d

where KNN(x,j) is the distance from x to its j-th nearest neighbor in Ci.

The mutual reachability distance between two objects xi and xj is:

MRD(xi,xj)=max{APCD(oi),APCD(xj),d(xi,xj)}

where d(xi,xj) is the Euclidean distance between xi and xj.

The minimum spanning tree (MST) is built for each cluster using mutual reachability distances. The density sparseness of cluster Ci can be defined as:

DSC(Ci)=max{weightofinternaledgesinMSTofCi}

The density separation between clusters Ci and Cj ( i≠j) is the following:

DSPC(Ci,Cj)=min{MRD(xp,xq)∣xp∈Ci,xq∈Cj}

And the resulting cluster validity score for Ci is:

VC(Ci)=minj≠iDSPC(Ci,Cj)−DSC(Ci)max(minj≠iDSPC(Ci,Cj),DSC(Ci))

Finally, the overall DBCV index for the full clustering solution can be summarized as:

(1) DBCV(C)=∑i=1k|Ci||X|⋅VC(Ci)

(minimum and worst value: −1; maximum and best value: +1).

Values near +1 indicate well-separated, cohesive clusters (in density terms), values around 0 suggest ambiguous cluster structure, values near −1 indicate poor clustering structure or misassignments.

Software packages

The original implementation of DBCV was in MATLAB (Jaskowiak, 2023), that unfortunately is a proprietary software: only users who buy a license can actually use it.

Fortunately, several versions of the DBCV index function are openly available on Python and R. In R, a function implementing the DBCV index recently published in the DBCVindex CRAN package (Chicco, 2025) and in the dbscan CRAN package (Hahsler). In Python, several packages reimplementing the original MATLAB version are available: FelSiq/DBCV (Siqueira, 2023), permetrics (van Thieu, 2024), k-DBCV (Kaufman Lab Columbia, 2024), and christopherjenness/DBCV (Jenness, 2023).

We tested the DBCV functions of all these software packages and found out that DBCVindex in R and FelSiq/DBCV in Python generated the most similar results to the original MATLAB implementation in reasonable time. We therefore decided to employ FelSiq/DBCV for this study.

The authors of CDbw and VIASCKDE released Python packages of their indices (Lashkov, 2019; Senol, 2024). DCSI was released as an R package (Gauss, 2023), and we used it in our Python environment through the rpy2 Python library (Gautier, 2025).

Clustering methods

To evaluate the reliability of the DBCV index and other metrics, we applied DBSCAN (Schubert et al., 2017) to artificial datasets, as detailed in the following sections. DBSCAN is a clustering algorithm that groups together points located in dense regions while identifying those in sparse areas as noise. It operates based on two main parameters: epsilon ( ϵ) and minimal points. The ϵ parameter defines the maximum distance within which points are considered part of the same cluster. A larger ϵ value tends to merge more points into clusters, whereas a smaller ϵ can result in more fragmented clusters or isolated noise points. The minimal points parameter sets the minimum number of neighboring points required to form a cluster. A point is categorized as a core point if it has at least minimal points within its ϵ-radius. Border points are those that are close to a core point but do not meet the minimal points threshold themselves, while noise points do not belong to any cluster. By carefully selecting ϵ and minimal points, DBSCAN is capable of detecting clusters of various shapes and managing outliers effectively.

Regarding the real-world medical data, we also employed HDBSCAN (McInnes, Healy & Astels, 2017) and MeanShift (Cheng, 1995). HDBSCAN is an advanced extension of DBSCAN that enhances cluster detection in datasets with variable density. Unlike DBSCAN, HDBSCAN builds a hierarchy of clusters based on density and then extracts the most stable partition automatically, making it more flexible and suitable for complex data.

Two key parameters of HDBSCAN are minimum cluster size and epsilon. The minimum cluster size parameter defines the minimum number of points required for a group to be considered a valid cluster. Higher values reduce the number of detected clusters and increase robustness against noise, while lower values allow the identification of smaller clusters. The epsilon parameter sets a distance threshold for separating clusters. If specified, this value ensures that two clusters are not merged if they are separated by a distance greater than ϵ, providing finer control over clustering granularity.

MeanShift is a density-based clustering algorithm that iteratively shifts data points toward regions of higher density until convergence is reached. Unlike HDBSCAN, it does not require specifying the number of clusters beforehand and can identify arbitrarily shaped structures, making it useful for exploratory data analysis.

A fundamental parameter of MeanShift is bandwidth, which defines the width of the search window used to estimate local density. A higher bandwidth value results in larger and more generalized clusters, whereas a lower value allows for the detection of more detailed clusters but may increase noise and over-segmentation. Choosing the optimal bandwidth is crucial for obtaining meaningful clustering results.

Validation on artificial data clustering trends

To validate the DBCV index on concave clusters, we generated four distinct artificial datasets, each designed to model specific geometries and clustering challenges. We intentionally designed these datasets as compositions of different clusters clearly observable by anyone, applied DBSCAN, and measured its results through seven internal clustering metrics: the DBCV index (Davoud et al., 2014), Silhouette coefficient (Rousseeuw, 1987), Dunn index (Dunn, 1974), Davies-Bouldin index (Davies & Bouldin, 1979), Calinski-Harabasz index (Calinski & Harabasz, 1974), and Shannon Entropy (Shannon, 1948). We then manipulated the data by inserting noise so that these clusters would become almost undistinguishable, reapplied DBSCAN, and reassessed its results through the same metrics. This noise data manipulation worsened the clustering results, and we expected to see the metrics to produce worsening results, too. However, not all the metrics confirmed this worsening.

Here we list the artificial data shapes that we employed in these tests.

Half Moons: The moon-shaped datasets were generated using a function, which utilizes the make_moons utility from the scikit-learn (Scikit Learn, 2007) Python library. This function creates datasets with two interlocking crescent shapes. We used 1,000 points with different noise levels (0.0, 0.056, 0.111, 0.167, 0.222, 0.278, 0.333, 0.389, 0.444, 0.5).

For each noise level, a separate dataset was created. Increasing the noise progressively blurs the crescent shapes, making the clustering task more challenging.

Shifting Circles: It is used a function to generate datasets consisting of an outer circle and a shifted inner circle. This design emphasizes the spatial relationship between the clusters. We utilized 10 datasets, having 500 points for the outer circle and 500 points for the inner circle. We set 0.5 as inner radius and 1.5 as outer radius. The initial shift was (0, 0) and the incremental shift was (0.2, 0).

The inner circle is progressively shifted along the x-axis with each dataset by the incremental shift value. This systematic displacement introduces varying degrees of overlap between the two circles, ranging from fully separate to partially overlapping clusters. These datasets serve as a valuable testbed for evaluating clustering algorithms’ capability to discern subtle structural changes while maintaining the integrity of distinct clusters. By incrementally adjusting the shift, we simulate scenarios where the boundary between clusters becomes increasingly ambiguous, pushing the limits of clustering methods.

Sparse Circles: Noisy concentric circle datasets were created using a function, which utilizes the make_circles utility from scikit-learn (Scikit Learn, 2007). The function generates two concentric circles with added Gaussian noise. Here we used ten datasets having 1,000 samples and by using a noise increment of 0.0555.

The noise level increases progressively across datasets, starting from a base value of 0.05. This variation tests clustering algorithms’ robustness to noise while maintaining the underlying concentric circle structure.

Tulip shapes: A special fourth dataset was generated to further assess the DBCV index. This dataset builds on existing crescent-shaped data, incorporating additional points and noise to create a more challenging test case. The original data was obtained from the moon example of the CVIK MATLAB package (José-García & Gómez-Flores, 2023; José-García). The process involved the following steps: 1. The Moon.mat file was loaded, and the x and y coordinates were extracted and combined into a single array of points.

2. The original points were normalized to lie within the [0, 1] range by scaling them based on their minimum and maximum values.

3. Augmentation: We generated 10 datasets, and each dataset included: • 400 points sampled from the normalized original points.

• Gaussian noise added to these new points to keep them close to the original distribution. Noise levels ranged from 0.0 to 0.5 in steps of 0.056.

4. The noisy points were wrapped within the [0, 1] range using the modulo operation.

5. The noisy new points were combined with the original normalized points to create the final augmented datasets.

This augmentation strategy effectively preserves the original distribution while introducing variability, thereby challenging clustering algorithms to identify clusters amidst increased noise and data density variations.

The rationale beyond this validation algorithm is simple: as the trends worsen, we expected a consistent deterioration of the internal clustering metrics’ results. If the results of a specific metric worsened, we considered that metric reliable. If the results stayed stable or improved, we considered that metric unreliable.

Validation on real-world medical data clustering trends

Clinical records data play a crucial role in medical research, offering valuable insights for patient care, disease progression analysis, and predictive modeling. These datasets typically consist of unstructured information, including demographic details, laboratory results, diagnostic codes, treatment histories, and physician notes.

In this study, we utilize five publicly available clinical datasets derived from health records spanning different diseases: Neuroblastoma (Ma et al., 2018), Diabetes type one (Takashi et al., 2019; Cerono & Chicco, 2024), Sepsis & SIRS (Gucyetmez & Atalan, 2016; Mollura et al., 2024), Heart Failure and Depression (Jani et al., 2016) and Cardiac Arrest (Requena-Morales et al., 2017).

The first dataset concerns neuroblastoma (Ma et al., 2018), a malignant tumor of the sympathetic nervous system. This disease is characterized by high genetic and clinical heterogeneity, making the analysis of multiple variables crucial for accurate diagnosis and risk stratification. The dataset includes 168 patients and 13 clinical features.

The second dataset relates to diabetes (Takashi et al., 2019; Cerono & Chicco, 2024), a chronic metabolic disorder characterized by elevated blood glucose levels due to insufficient insulin production or utilization. Since diabetes is associated with numerous complications, the dataset includes 67 patients and 20 clinical.

Another dataset focuses on sepsis and Systemic Inflammatory Response Syndrome (SIRS) (Gucyetmez & Atalan, 2016; Mollura et al., 2024). Sepsis is a systemic inflammatory response triggered by a severe infection, whereas SIRS can occur without an infectious agent. Differentiating between these conditions is crucial for timely medical intervention. This dataset contains data from 1,257 patients with 16 clinical features relevant for identifying and managing these syndromes.

We also analyzed a dataset on heart failure and depression (Jani et al., 2016). Heart failure is a chronic condition in which the heart cannot pump blood efficiently, leading to inadequate tissue oxygenation. Depressive symptoms are common in these patients and can negatively impact prognosis. The dataset includes 425 patients with 15 clinical features to explore this complex relationship.

Finally, the dataset on cardiac arrest (Requena-Morales et al., 2017) provides information on 420 patients and 10 clinical features. Cardiac arrest is a sudden and potentially fatal event characterized by the cessation of cardiac activity. Analyzing this dataset is essential for identifying risk factors and improving emergency interventions.

We used datasets form electronic health records because of their high level of complexity. These data, in fact, were collected for clinical reasons and not for scientific purposes, through standard hospital laboratory tests (such as blood test). Moreover, these data consists of different data types (numeric, ordinal, categorical, binary) making their clustering particularly challenging. Nevertheless, machine learning and computational statistics applied to these data can be extremely useful for predicting prognosis or disease trends of patients (Patton & Liu, 2023; Steele et al., 2018; Wong et al., 2018).

Since these datasets are real-world clinical records with complex and heterogeneous structures, assessing the reliability of a clustering validation metric like DBCV becomes challenging. Unlike artificial datasets with well-defined clusters, medical data often contain overlapping patient profiles, missing values, and noisy features, making it difficult to determine a “ground truth” for clustering quality. The effectiveness of DBCV in such scenarios depends on its ability to capture meaningful patient groupings despite these complexities. Therefore, while DBCV provides valuable insights, its performance should be interpreted alongside other validation indices and domain knowledge to ensure clinically relevant results.

To assess the reliability of the DBCV metric, the proposed approach involves evaluating its stability as clustering parameters deteriorate. For each dataset, we selected a set of random hyperparameters for DBSCAN, HDBSCAN, and MeanShift. Each metric’s trend was then analyzed by comparing its values when transitioning from these random parameters to default settings. If the variation in the DBCV score aligns with the trend observed in the ARI, the metric is considered consistent; otherwise, it is deemed inconsistent.

We calculate the ARI values of the pairs: clusters generated by DBSCAN and clusters generated by HDBSCAN; clusters generated by DBSCAN and clusters generated by MeanShift; clusters generated by MeanShift and clusters generated by DBSCAN. We then calculate the average value of these three scores. We represent this process in Fig. 1.

Figure 1 Flowchart of the validation process on the medical datasets.

Average ARI: ((ARI between the results of DBSCAN and HDBSCAN) + (ARI between the results of HDBSCAN and MeanShift) + (ARI between the results of MeanShift and DBSCAN))/3. Internal clustering evaluation metric: each of the scores considered in this study, that are DBCV, DCSI, CDbw, and VIASCKDE.

In summary, if the clustering trends measured through ARI were worsening and the values of a specific metric were worsening too, we considered that metric reliable. The same thing for improvements: improving ARI trend and improving metric’s results made that metric reliable. In all the other cases, we considered the metric unreliable (Fig. 1).

Results

In this section, we first describe the results obtained on the artificial datasets (‘artificial clouds of points’) and then the results achieved on the medical datasets (‘clinical records data tests’).

Artificial clouds of points

The analysis involves generating multiple artificial datasets, each subjected to clustering using the DBSCAN algorithm (Ester et al., 1996). This method is particularly effective in identifying arbitrarily shaped clusters and managing noisy data.

For each dataset, three visualizations are produced: 1. Original dataset (left): A graphical representation of the generated points without clustering.

2. DBSCAN clustering (centre): A colored visualization illustrating how DBSCAN partitions the data into different clusters.

3. Evaluation metrics (right): A textual summary displaying various clustering validation indices.

Half moons

As observed in Figs. 2 and in 3, clustering performance progressively deteriorates with increasing noise levels. Initially, DBSCAN effectively identifies well-defined clusters, as reflected by high DBCV, VIASCKDE and DCSI scores. However, as noise intensifies, the quality of clustering declines, with a growing number of misclassified points and fragmented clusters. The number of detected clusters rises unpredictably, highlighting the algorithm’s sensitivity to noise. Additionally, the DBCV score steadily decreases, indicating a weakening of the density-based cluster structures. Among the validation metrics, DBCV initially aligns closely with the DCSI index, confirming well-defined clusters in low-noise conditions. However, as noise increases, DBCV deteriorates more gradually than the VIASCKDE score, which rapidly decrease, suggesting that DBCV is more robust in assessing density-based clustering structures. Interestingly, both VIASCKDE and DCSI exhibit a slight increase at the initial rise of noise, indicating a lower robustness of these metrics in the early stages of degradation.

Figure 2 Clustering results for the Half Moons dataset—upper part.

This image presents five datasets with increasing noise levels. For each dataset, we show the original data, the clusters obtained using the DBSCAN algorithm, and the corresponding clustering metric values. The noise level increases by 0.0555 for each dataset. The DBCV and VIASCKDE scores range within [−1;+1], while the DCSI Index have a range of [0, 1]. For all of them, higher values indicate better clustering quality. For CDbw, the higher the metric value, the better the clustering quality. The second part of this image can be found in Fig. 3.

Figure 3 Clustering results for the Half Moons dataset—lower part.

The first part of this image can be found in Fig. 2.

The trends observed in Fig. 4 provide key insights into the effect of noise on clustering performance. The DBCV score remains relatively high at low noise levels, indicating well-defined clusters, but progressively declines as noise increases. Despite this decline, DBCV remains above other metrics in its ability to evaluate density-based structures, showing a more gradual deterioration. The VIASCKDE index performs reliably in the initial stages, decreasing appropriately as noise levels rise during the first four evaluation steps. However, it unexpectedly begins to increase afterwards—despite the dataset becoming progressively noisier. This behavior indicates a potential weakness in the metric’s ability to distinguish degraded clustering quality in higher-noise scenarios.

Figure 4 Metrics on DBSCAN tests for Half Moons dataset.

These figures present the variation of the metrics computed in Figs. 2 and 3, as noise is added to the original plot. The value of each metric is plotted as noise increases.

The CDbw index behaves counterintuitively: it reports lower values when there is no noise, and increases almost proportionally with the noise level. This trend challenges the expectation that higher values always reflect better clustering, and suggests that CDbw may not effectively penalize the fragmentation caused by noise.

Similarly, the DCSI index initially shows a correct downward trend as noise increases, signaling reduced cluster separability and connectivity. However, it too enters a phase of growth at higher noise levels, deviating from what one would expect in a scenario of worsening cluster structure. This pattern, shared with VIASCKDE, points to a reduced robustness when facing significant data degradation.

Shifting circles

As observed in Figs. 5, 6 and 7, the clustering performance remains relatively stable across different datasets, despite the gradual displacement of the inner circle. DBSCAN consistently detects two clusters, as evidenced by the DBCV score, which remains around 0.5 throughout the datasets. This stability suggests that the algorithm effectively captures the density-based structure, even as the spatial arrangement of the clusters shifts.

Figure 5 Clustering results for the shifting circles dataset—upper part.

This image presents five datasets with increasing noise levels. For each dataset, we show the original data, the clusters obtained using the DBSCAN algorithm, and the corresponding clustering metric values. The noise level increases by 0.0555 for each dataset. The DBCV and VIASCKDE scores range within [−1;+1], while the DCSI has a range of [0,1]. For all of them, higher values indicate better clustering quality. For CDbw, the higher the metric value, the better the clustering quality. The second part of this figure can be found in Fig. 6.

Figure 6 Clustering results for the shifting circles dataset—lower part.

This image presents five datasets with increasing noise levels. For each dataset, we show the original data, the clusters obtained using the DBSCAN algorithm, and the corresponding clustering metric values. The noise level increases by 0.0555 for each dataset. The DBCV and VIASCKDE scores range within [−1;+1], while the DCSI has a range of [0,1]. For all of them, higher values indicate better clustering quality. For CDbw, the higher the metric value, the better the clustering quality. The first part of this figure can be found in Fig. 5.

Figure 7 Metrics on DBSCAN tests for the shifting circles dataset.

These figures present the variation of the metrics computed in Figs. 5 and 6, as noise is added to the original plot. The value of each metric is plotted as noise increases.

VIASCKDE remains relatively stable across the initial observations but shows a notable increase in its values when the two concentric circles begin to intersect. While the score appears consistent at first, it fails to maintain coherence with the actual cluster quality and does not accurately reflect good separation in the early, less noisy stages. This behavior suggests that VIASCKDE may lack sensitivity to subtle structural differences at low noise levels, and its sharp increase under higher noise may indicate an overestimation of clustering quality.

The CDbw index exhibits counterintuitive behavior throughout the evaluation. It starts with a very low value, then peaks unexpectedly, gradually declines, and eventually rises again. This non-monotonic trend diverges from the expected pattern of continuous degradation as noise increases. The observed increases may be due to a misinterpretation of dispersed noise points as inter-cluster density, artificially boosting the score. These fluctuations undermine the reliability of CDbw in scenarios involving progressive noise, suggesting it may not be well-suited for evaluating clustering quality in such contexts.

The DCSI score remains flat at zero across all evaluations, indicating no detected core structure or cluster separability according to its internal criteria. This persistent null response may stem from a conservative threshold for core point identification or a limited responsiveness to mild structural changes. Although DCSI may become more informative as the dataset degrades further, its lack of reaction in the early stages limits its practical utility under low-noise conditions.

Sparse circles

The trends observed in Figs. 8, 9 and 10 highlight the effects of increasing noise on clustering performance in sparse circular datasets.

Figure 8 Clustering results for the Sparse Circles dataset—upper part.

This image presents five datasets with increasing noise levels. For each dataset, we show the original data, the clusters obtained using the DBSCAN algorithm, and the corresponding clustering metric values. The noise level increases by 0.0555 for each dataset. The DBCV and VIASCKDE scores range within [−1;+1], while the DCSI Index have a range of [0, 1]. For all of them, higher values indicate better clustering quality. For CDbw, the higher the metric value, the better the clustering quality. The second part of this figure can be found in Fig. 9.

Figure 9 Clustering results for the Sparse Circles dataset—lower part.

This image presents five datasets with increasing noise levels. For each dataset, we show the original data, the clusters obtained using the DBSCAN algorithm, and the corresponding clustering metric values. The noise level increases by 0.0555 for each dataset. The DBCV and VIASCKDE scores range within [−1;+1], while the DCSI Index have a range of [0, 1]. For all of them, higher values indicate better clustering quality. For CDbw, the higher the metric value, the better the clustering quality. This figure is a continuation of Fig. 8. The first part of this figure can be found in Fig. 8.

Figure 10 Metrics on DBSCAN tests for Sparse Circles dataset.

These figures presents the variation of the metrics computed in Figs. 8 and 9, as noise is added to the original plot. The value of each metric is plotted as noise increases.

The DBCV, VIASCKDE, and DCSI indices all show an initial decrease as noise increases, suggesting that they are all sensitive—at least in the early stages—to the loss of clear cluster structure. This common trend highlights how even moderate levels of noise can begin to erode the cohesion and separation that define meaningful clustering.

After this initial decline, DBCV tends to stabilize, maintaining moderately low values with only minor fluctuations. While it no longer reflects a strong cluster structure, its relative steadiness suggests it still captures some residual density information, even as noise increases.

VIASCKDE, on the other hand, displays a more irregular pattern after its initial drop. Its values begin to oscillate noticeably, alternating between rises and falls across different noise levels. This behavior might reflect a heightened sensitivity to local density changes, but also hints at instability—perhaps overreacting to the introduction of scattered noise rather than robustly tracking meaningful structure.

The DCSI index follows a similar fluctuating trend but does so more gently. Its variations are less pronounced, indicating a more conservative response to noise. While it does react to changes in the data, its limited range of movement suggests that it might miss more subtle transitions in cluster connectivity or underrepresented mild structural degradation.

In contrast, the CDbw index behaves differently. Instead of decreasing or oscillating, it gradually increases as noise rises. This unexpected trend could be due to the metric misinterpreting dispersed noise points as contributing positively to inter-cluster density, which artificially inflates its value.

Tulip dataset

In this scenario, all the evaluated metrics exhibit a consistent trend, showing similar behavior as noise increases Figs. 11 and 12. Despite some local fluctuations, the overall trajectory for each metric is downward, indicating a progressive degradation in clustering quality as noise disrupts the underlying data structure (Fig. 13).

Figure 11 Clustering results for the Tulip dataset—upper part.

This image presents five datasets with increasing noise levels. For each dataset, we show the original data, the clusters obtained using the DBSCAN algorithm, and the corresponding clustering metric values. The noise level increases by 0.0555 for each dataset. The DBCV and VIASCKDE scores range within [−1;+1], while the DCSI Index have a range of [0, 1]. For all of them, higher values indicate better clustering quality. For CDbw, the higher the metric value, the better the clustering quality. The second part of this figure can be found in Fig. 12.

Figure 12 Clustering results for the Tulip dataset—lower part.

This image presents five datasets with increasing noise levels. For each dataset, we show the original data, the clusters obtained using the DBSCAN algorithm, and the corresponding clustering metric values. The noise level increases by 0.0555 for each dataset. The DBCV and VIASCKDE scores range within [−1;+1], while the DCSI Index have a range of [0, 1]. For all of them, higher values indicate better clustering quality. For CDbw, the higher the metric value, the better the clustering quality. The first part of this figure can be found in Fig. 11.

Figure 13 Metrics on DBSCAN tests for Tulip dataset.

These figures present the variation of the metrics computed in Figs. 11 and 12, as noise is added to the original plot. The value of each metric is plotted as noise increases.

Interestingly, even the CDbw index—which in previous cases displayed an opposite trend by increasing with noise—aligns with the behavior of the other metrics in this setting, showing a general decline. While some oscillations are still present and occasionally interrupt the monotonic descent, they do not significantly alter the overall pattern. These irregularities likely reflect residual sensitivity to local noise effects or internal variations in cluster structure, but the dominant trend remains one of decreasing cluster cohesion and separability in response to increasing noise.

Recap of the results on the artificial data tests. To assess the robustness of clustering validation metrics under increasing noise levels, a consistency criterion was established. A metric is deemed consistent if its values predominantly decrease as the noise increases, allowing for minor fluctuations.

The method involves a systematic analysis of each metric’s behavior, by counting the number of consecutive value pairs that decrease.

Metrics are classified as inconsistent if their values predominantly increase with higher noise levels, or if they remain constant across all noise levels—since such invariance suggests a lack of sensitivity to structural changes in the data.

Table 2 provides a summary of the consistency of various clustering validation metrics across different datasets. Each row corresponds to a specific metric, while the columns represent the datasets: Half Moons, Shifting Circles, Sparse Circles, and Tulip. The values in the cells indicate the number of times a given metric was found to be consistent within each dataset. The final column (Total) reports the overall number of consistent outcomes observed across all datasets as noise increased.

Table 2 Metric decreasing for different datasets.

The table presents a summary of metric decreasing across all datasets (Half Moons, Shifting Circles, Sparse Circles, and Tulip).

Metric	Half moons	Shifting circles	Sparse circles	Tulip	Total	
DBCV	7	7	5	7	26	
VIASCKDE	5	6	5	4	20	
CDbw	3	5	2	6	16	
DCSI	5	0	3	6	14	

From the table, it is evident that DBCV is the most stable metric, showing the highest number of consistent trends both overall and within each individual dataset. Its values remained relatively stable and consistently outperformed the other metrics across noise levels.

VIASCKDE appears to follow closely, ranking as the second most consistent metric. In contrast, CDBW and DCSI exhibit limited stability, indicating a weaker responsiveness to changes in data structure under increasing noise.

Clinical records data tests

For each of the five datasets, a table is provided displaying the values of the various metrics, indicating whether the trend of the metric is consistent with that of the ARI. Additionally, a second table summarizes the frequency with which each metric is classified as consistent or inconsistent across the datasets.

Neuroblastoma dataset. We applied our validation approach described in subsection ‘Validation on real-world medical data clustering trends’ to a dataset derived from health records of patients with neuroblastoma (Ma et al., 2018).

We measured the trend of the ARI when moving from tests made through random parameters to tests made through default parameters (Table 3). The results showed that DBCV index resulted being consistent with the ARI trends on all three tests, followed by DCSI index and VIASCKDE on 2 tests (Table 4).

Table 3 Comparison of metrics across methods: neuroblastoma.

This table compares metric performance under random hyperparameters vs default settings for different clustering methods, highlighting the observed trends and their consistency with the ARI metric on the neuroblastoma dataset. The ARI value for the random parameters is 0.345 while for the default parameters is 0.135. For HDBSCAN, the random parameters are minimum cluster size = 12 and epsilon = 0.854, while the default parameters settings are minimum cluster size = 5 and epsilon = 0.0. For DBSCAN, the random parameters are minimal points = 3 and epsilon = 0.274, whereas the default parameters settings are epsilon = 0.5 and minimal points = 5. For MeanShift, the random bandwidth is selected randomly by picking a real value between 0.1 and 5, while the default parameters configuration bandwidth is estimated using the bandwidth estimation function of scikit-learn.

Metric	Random parameters value	Default parameters value	Clustering method	Trend	Trend consistency with ARI	
DBCV	0.15	0.06	DBSCAN	↘	Consistent	
VIASCKDE	0.13	0.15	DBSCAN	↗	Inconsistent	
CDbw	–	–	DBSCAN	–	Not applicable	
DCSI	0.66	0.89	DBSCAN	↗	Inconsistent	
DBCV	0.47	0.42	HDBSCAN	↘	Consistent	
VIASCKDE	0.14	0.12	HDBSCAN	↘	Consistent	
CDbw	–	–	HDBSCAN	–	Not applicable	
DCSI	0.56	0.55	HDBSCAN	↘	Consistent	
DBCV	–0.82	–0.83	MeanShift	↘	Consistent	
VIASCKDE	0.13	0.13	MeanShift	↘	Consistent	
CDbw	–	–	MeanShift	–	Not applicable	
DCSI	0.22	0.22	MeanShift	↘	Consistent	

Table 4 Trend consistency with ARI: neuroblastoma.

This table summarizes the number of methods for each metrics that are consistent or inconsistent with the ARI for the neuroblastoma dataset.

	Neuroblastoma dataset	
Metric	# Consistent	% Consistent	# Inconsistent	# Not applicable	
DBCV	3	100%	0	0	
DCSI	2	67%	1	0	
VIASCKDE	2	67%	1	0	
CDbw	0	0%	0	3	

Diabetes type one dataset. We applied our validation approach described in ‘Validation on real-world medical data clustering trends’ to a dataset derived from health records of patients with diabetes type (Takashi et al., 2019; Cerono & Chicco, 2024).

We measured the trend of the Adjusted Rand Index when moving from tests made through random parameters to tests made through default parameters (Table 5). The results showed that no index was consistent with the ARI trend for all the tests made: DCSI index resulted consistent on 2 tests out of 3, while DBCV index only once (Table 6).

Table 5 Comparison of metrics across methods: diabetes.

This table compares metric performance under random hyperparameters vs default settings for different clustering methods, highlighting the observed trends and their consistency with the ARI metric on the diabetes dataset. The ARI value for the random parameters is 0.201 while for the default parameters is 0.153. For HDBSCAN, the random parameters are minimum cluster size = 13 and epsilon = 0.744, while the default parameters settings are minimum cluster size = 5 and epsilon = 0.0. For DBSCAN, the random parameters are minimal points = 6 and epsilon = 1.181, whereas the default parameters settings are epsilon = 0.5 and minimal points = 5. For MeanShift, the random bandwidth is selected randomly by picking a real value between 0.1 and 5, while the default parameters configuration bandwidth is estimated using the bandwidth estimation function of scikit-learn.

Metric	Random parameters value	Default parameters value	Clustering method	Trend	Trend consistency with ARI	
DBCV	0.47	0.00	DBSCAN	↘	Consistent	
VIASCKDE	–	–	DBSCAN	–	Not applicable	
CDbw	–	–	DBSCAN	–	Not applicable	
DCSI	0.53	–	DBSCAN	–	Not applicable	
DBCV	0.33	0.47	HDBSCAN	↗	Inconsistent	
VIASCKDE	–	–	HDBSCAN	–	Not applicable	
CDbw	–	–	HDBSCAN	–	Not applicable	
DCSI	0.60	0.53	HDBSCAN	↘	Consistent	
DBCV	–0.56	0.03	MeanShift	↗	Inconsistent	
VIASCKDE	–	–	MeanShift	–	Not applicable	
CDbw	–	–	MeanShift	–	Not applicable	
DCSI	0.24	0.23	MeanShift	↘	Consistent	

Table 6 Trend consistency with ARI: Diabetes.

This table summarizes the number of methods for each metrics that are consistent or inconsistent with the adjusted rand index (ARI) for the diabetes dataset.

	Diabetes dataset	
Metric	# Consistent	% Consistent	# Inconsistent	# Not applicable	
DCSI	2	67%	0	1	
DBCV	1	33%	2	0	
CDbw	0	0%	0	3	
VIASCKDE	0	0%	0	3	

Sepsis and SIRS dataset. We applied our validation approach described in ‘Validation on real-world medical data clustering trends’ to a dataset derived from health records of patients with sepsis or SIRS (Gucyetmez & Atalan, 2016; Mollura et al., 2024).

We measured the trend of the Adjusted Rand Index when moving from tests made through random parameters to tests made through default parameters (Table 7). The results showed that DBCV index resulted informative on two tests, while DCSI index on one tests out of three (Table 8).

Table 7 Comparison of metrics across methods: Sepsis and SIRS.

This table compares metric performance under random hyperparameters vs default parameters for different clustering methods, highlighting the observed trends and their consistency with the ARI metric on the Sepsis and SIRS dataset. The ARI value for the random parameters is 0.612 while for the default parameters is 0.348. For HDBSCAN, the random parameters are minimum cluster size = 13 and epsilon = 0.423, while the default parameters settings are minimum cluster size = 5 and epsilon = 0.0. For DBSCAN, the random parameters are minimal points = 4 and epsilon = 0.859, whereas the default parameters settings are epsilon = 0.5 and minimal points = 5. For MeanShift, the random bandwidth is selected randomly by picking a real value between 0.1 and 5, while the default parameters configuration bandwidth is estimated using the bandwidth estimation function of scikit-learn.

Metric	Random parameters value	Default parameters value	Clustering method	Trend	Trend consistency with ARI	
DBCV	–0.79	0.32	DBSCAN	↗	Inconsistent	
VIASCKDE	–	0.13	DBSCAN	–	Not applicable	
CDbw	–	–	DBSCAN	–	Not applicable	
DCSI	0.08	0.36	DBSCAN	↗	Inconsistent	
DBCV	0.37	0.36	HDBSCAN	↘	Consistent	
VIASCKDE	0.07	0.11	HDBSCAN	↗	Inconsistent	
CDbw	–	–	HDBSCAN	–	Not applicable	
DCSI	0.40	0.37	HDBSCAN	↘	Consistent	
DBCV	0.19	–0.81	MeanShift	↘	Consistent	
VIASCKDE	–	0.09	MeanShift	–	Not applicable	
CDbw	–	–	MeanShift	–	Not applicable	
DCSI	–	0.17	MeanShift	–	Not applicable	

Table 8 Trend consistency with ARI: Sepsis and SIRS.

This table summarizes the number of methods for each metrics that are consistent or inconsistent with the adjusted rand index (ARI) for the sepsis & SIRS dataset.

	Sepsis and SIRS dataset	
Metric	# Consistent	% Consistent	# Inconsistent	# Not applicable	
DBCV	2	67%	1	0	
DCSI	1	33%	1	1	
CDbw	0	0%	0	3	
VIASCKDE	0	0%	1	2	

Heart Failure and Depression dataset. We applied our validation approach described in ‘Validation on real-world medical data clustering trends’ to a dataset derived from health records of the heart failure and depression among patients (Jani et al., 2016).

We measured the trend of the ARI when moving from tests made through random parameters to tests made through default parameters (Table 9). The results showed no test obtained consistent trends with the ARI trends on all the three tests (Table 10).

Table 9 Comparison of metrics across methods: Heart Failure and Depression.

This table compares metric performance under random hyperparameters vs default parameters settings for different clustering methods, highlighting the observed trends and their consistency with the ARI metric on the heart failure and depression dataset. The ARI value for the random parameters is 0.314 while for the default parameters is 0.3 For HDBSCAN, the random parameters are minimum cluster size = 18 and epsilon = 0.524, while the default parameters settings are minimum cluster size = 5 and epsilon = 0.0. For DBSCAN, the random parameters are minimal points = 15 and epsilon = 1.08, whereas the default parameters settings are epsilon = 0.5 and minimal points = 5. For MeanShift, the random bandwidth is selected randomly by picking a real value between 0.1 and 5, while the default parameters configuration bandwidth is estimated using the bandwidth estimation function of scikit-learn.

Metric	Random parameters value	Default parameters value	Clustering method	Trend	Trend consistency with ARI	
DBCV	0.35	0.42	DBSCAN	↗	Inconsistent	
VIASCKDE	0.04	0.05	DBSCAN	↗	Inconsistent	
CDbw	–	–	DBSCAN	–	Not applicable	
DCSI	0.60	0.67	DBSCAN	↗	Inconsistent	
DBCV	0.14	0.56	HDBSCAN	↗	Inconsistent	
VIASCKDE	0.01	0.08	HDBSCAN	↗	Inconsistent	
CDbw	–	–	HDBSCAN	–	Not applicable	
DCSI	0.50	0.49	HDBSCAN	↘	Consistent	
DBCV	0.21	–0.73	MeanShift	↘	Consistent	
VIASCKDE	–	0.01	MeanShift	–	Not applicable	
CDbw	–	–	MeanShift	–	Not applicable	
DCSI	–	0.20	MeanShift	–	Not applicable	

Table 10 Trend consistency with ARI: Heart Failure and Depression.

	Heart failure and depression dataset	
Metric	# Consistent	% Consistent	# Inconsistent	# Not applicable	
DBCV	1	33%	2	0	
DCSI	1	33%	1	1	
CDbw	0	0%	0	3	
VIASCKDE	0	0%	2	1	
Note:

This table (Table 9) summarizes the number of methods for each metrics that are consistent or inconsistent with the adjusted rand index (ARI) for the heart failure and depression dataset.

Cardiac arrest dataset. We applied our validation approach described in ‘Validation on real-world medical data clustering trends’ to a dataset derived from health records of patients with cardiac arrest (Requena-Morales et al., 2017).

We measured the trend of the Adjusted Rand Index when moving from tests made through random parameters to tests made through default parameters (Table 11). The results showed no test obtained consistent trends with the ARI trends on all the three tests (Table 12).

Table 11 Comparison of metrics across methods: cardiac arrest.

This table compares metric performance under random hyperparameters vs default parameters settings for different clustering methods, highlighting the observed trends and their consistency with the ARI metric on the heart failure and depression dataset. The ARI value for the random parameters is 0.330 while for the default parameters is 0.472. For HDBSCAN, the random parameters are minimum cluster size = 12 and epsilon = 0.310, while the default parameters settings are minimum cluster size = 5 and epsilon = 0.0. For DBSCAN, the random parameters are minimal points = 9 and epsilon = 0.676, whereas the default parameters settings are epsilon = 0.5 and minimal points = 5. For MeanShift, the random bandwidth is selected randomly by picking a real value between 0.1 and 5, while the default parameters configuration bandwidth is estimated using the bandwidth estimation function of scikit-learn.

Metric	Random parameters value	Default parameters value	Clustering method	Trend	Trend consistency with ARI	
DBCV	0.46	0.64	DBSCAN	↗	Consistent	
VIASCKDE	0.23	0.28	DBSCAN	↗	Consistent	
CDbw	–	–	DBSCAN	–	Not applicable	
DCSI	0.60	0.67	DBSCAN	↗	Consistent	
DBCV	0.35	0.61	HDBSCAN	↗	Consistent	
VIASCKDE	0.19	0.23	HDBSCAN	↗	Consistent	
CDbw	–	–	HDBSCAN	–	Not applicable	
DCSI	0.50	0.50	HDBSCAN	↘	Inconsistent	
DBCV	0.47	–0.28	MeanShift	↘	Inconsistent	
VIASCKDE	–	0.24	MeanShift	–	Not applicable	
CDbw	–	–	MeanShift	–	Not applicable	
DCSI	–	0.05	MeanShift	–	Not applicable	

Table 12 Trend consistency with ARI: cardiac arrest.

This table summarizes the number of methods for each metrics that are consistent or inconsistent with the adjusted rand index (ARI) for the cardiac arrest dataset.

	Cardiac arrest dataset	
Metric	# Consistent	% Consistent	# Inconsistent	# Not applicable	
DBCV	2	67%	1	0	
VIASCKDE	2	67%	0	1	
DCSI	1	50%	1	1	
CDbw	0	0%	0	3	

Recap of the medical data tests. We counted the number of times each clustering internal metric had consistent trend compared with the ARI trend (Fig. 14). Since we employed three clustering techniques (DBSCAN, HDBSCAN, and MeanShift) on five medical datasets (Neuroblastoma, Sepsis and SIRS, Heart Failure and Depression, Diabetes type one, Cardiac Arrest), we eventually had 15 tests for each metric. Our results indicated that the DBCV index is the most informative metric among the seven scores studied, by having a consistent trend on nine cases our of 15 (around 60%). The DCSI index obtained a 46.7% success rate, by resulting informative on seven datasets out of 15. VIASCKDE were coherent with the ARI trend on four times, while CDbw is not applicable to any of the datasets. These results prove that the DBCV index is more informative than these other clustering metrics considered in this study.

Figure 14 Recap of all the trend consistencies.

The barchart summarizes the number of consistent and inconsistent trends with the adjusted rand index (ARI) across all the 15 tests made on all the datasets. DBCV index: nine times. DCSI index: seven times. VIASCKDE index: four times. CDbw index: zero times.

Discussion and conclusions

Assessing the results of a clustering analysis can be difficult, because we have no gold standard labels telling us if our machine learning technique performed well or not: this aspect is they key problem of unsupervised machine learning approaches. Clustering results can be evaluated externally, through the ARI for example, when some “real clusters” are available for a comparison, somehow framing the scientific problem into a supervised machine learning approach (Jaskowiak, Costa & Campello, 2022). But when no cluster true information is available, internal clustering scores are needed to evaluate the quality of the results obtained.

Several internal clustering indices exist, such as Silhouette coefficient, Calinski-Harabasz index, Davies-Bouldin, Dunn index, Gap statistic, and Shannon entropy, just to mention a few that we commonly saw in our biomedical informatics research experience. These metrics work well when the clusters are clearly separable and convex-shaped, but can be misleading when the clusters are concave or nested. In these cases, a density-based score is needed, and the DBCV index by Davoud et al. (2014) was proposed to solve this problem.

The original DBCV study by Davoud et al. (2014) proposed a short comparison between DBCV and SWC, VRC, Dunn index, the MB index, and CDbw, and performed their tests only on artificial data. In this study, instead, we compared the behavior of the DBCV index again with CDbw and with other coefficients designed for concave, nested clusters (DCSI and VIASCKDE), through extensive tests on artificial data and on medical data derived from electronic health records.

To do so, we proposed a novel validation approach based on artificial data representing clear shapes that deteriorate by adding noise points and a novel validation procedure based on comparing results obtained through different configurations of hyperparameters on real-world Electronic Health Records (EHRs) datasets. Our results confirm the effectiveness of the DBCV index not only on artificial data, but also on real-world medical datasets, and the higher informativeness of this metric over the other six scores considered.

Several studies on clustering internal metrics exist, but most of them rely on finding the “best” number of clusters for several applications: we decided not to participate in this never-ending debate but rather focus on data cases where clustering results clearly worsen or improve, and see if the considered metrics confirmed these outcome trends or not. Our innovative approach proved that the DBCV index can produce results which are consistent with the expected and observed clustering outcomes.

Like in any scientific work, our study presents some limitations, too. Even if selected artificial data clouds of points clearly separated as concave-shaped clusters, we cannot affirm that all the clusters found by our methods on the medical data are concave or nested, since they are in the N-dimensional space. In the future, we plan to compare the DBCV index with other pupular metrics such as the Within-Cluster Sum of Squares (WCSS) (Brusco & Stahl, 2005), the Xie-Beni index clustering (Xie & Beni, 1991), or the Hartigan index (Hartigan, 1972).

Another limitation of our study is that we employed only medical data from electronic health records, and no data of other types such as medical images or physiologic signs. We decided to use EHRs because these data are complex, mixed-types (numerical, ordinal, categorical, etc.), and they were not collected for scientific purposes. The level of complexity in clustering EHRs data is high, and therefore a clustering analysis on datasets of this type can be considered valid and robust. In the future, we plan to apply our approach to biomedical data of other types too.

In this study, we decided to employ only density-based clustering coefficients that are available as open source packages in Python or R, following the principles of open science for reproducibility (Sandve et al., 2013). We advocate for the use of only open-source programming languages and software.

To conclude, we recommend choosing the DBCV index over DCSI, CDbw, and VIASCKDE for assessing any clustering analysis producing density-based, concave-shaped, or nested clusters.

Ethics approval and consent to participate The permissions to collect and analyze the data of patients’ involved in this study were obtained by the original dataset curators (Gucyetmez & Atalan, 2016; Mollura et al., 2024; Jani et al., 2016; Requena-Morales et al., 2017; Ma et al., 2018; Takashi et al., 2019; Cerono & Chicco, 2024).

The authors utilized Ecosia AI Chat for English proof-reading the text of the manuscript.

List of abbreviations

ARI Adjusted Rand index

BIRCH balanced iterative reducing and clustering using hierarchies

CC Creative Commons

CDbw Compose Density between and within clusters

CRAN Comprehensive R Archive Network

DBCV density-based clustering validation

DBI Davies-Bouldin index

DBSCAN density-based spatial clustering of applications with noise

DCSI density cluster separability index

EHRs electronic health records

GPL GNU General Public License

HDBSCAN hierarchical density-based spatial clustering of applications with noise

MB Maulik-Bandyopadhyay index

NB neuroblastoma

PGS polygenic scores

SIRS systemic inflammatory response syndrome

SWC Silhouette Width Criterion

UMAP Uniform Manifold Approximation and Projection

URL uniform resource locator

VIASCKDE Validity Index for Arbitrary-Shaped Clusters based on the kernel density estimation

VRC Variance Ratio Criterion.

Additional Information and Declarations

Competing Interests

Davide Chicco is an Academic Editor for PeerJ Computer Science.

Author Contributions

Davide Chicco conceived and designed the experiments, analyzed the data, prepared figures and/or tables, authored or reviewed drafts of the article, and approved the final draft.

Giuseppe Sabino performed the experiments, performed the computation work, prepared figures and/or tables, authored or reviewed drafts of the article, and approved the final draft.

Luca Oneto analyzed the data, authored or reviewed drafts of the article, and approved the final draft.

Giuseppe Jurman analyzed the data, authored or reviewed drafts of the article, and approved the final draft.

Data Availability

The following information was supplied regarding data availability:

The Python code developed and used in this study is available at GitHub and Zenodo:

- https://github.com/GiuseppeSabino01/DBCV-paper, GPL-3.0 license.

- Giuseppe Sabino. (2025). GiuseppeSabino01/DBCV-paper: The DBCV index is more informative than DCSI, CDbw, and VIASCKDE indices for unsupervised clustering internal assessment of concave-shaped and density-based clusters (v1.0.0). Zenodo. https://doi.org/10.5281/zenodo.15710281.

– The Cardiac Arrest dataset is available at figshare: Requena-Morales, Rosa; Palazón-Bru, Antonio; Mercedes Rizo-Baeza, María; Adsuar-Quesada, José Manuel; Francisco Gil-Guillén, Vicente; Cortés-Castell, Ernesto (2017). Database of our study. PLOS ONE. Dataset. https://doi.org/10.1371/journal.pone.0175818.s001.

– The Diabetes type one dataset is available at figshare: Takashi, Yuichi; Ishizu, Masashi; Mori, Hiroyasu; Miyashita, Kazuyuki; Sakamoto, Fumie; Katakami, Naoto; et al. (2019). Circulating osteocalcin as a bone-derived hormone is inversely correlated with body fat in patients with type 1 diabetes. PLOS ONE. Dataset. https://doi.org/10.1371/journal.pone.0216416.

– The Heart Failure and Depression dataset is available at figshare: Dinesh Jani, Bhautesh; S. Mair, Frances; Roger, Véronique L.; Weston, Susan A.; Jiang, Ruoxiang; M. Chamberlain, Alanna (2016). Comorbid Depression and Heart Failure: A Community Cohort Study. PLOS ONE. Dataset. https://doi.org/10.1371/journal.pone.0158570.

– The Neuroblastoma dataset is available at PeerJ: https://doi.org/10.7717/peerj.5665/supp-5.

– The Sepsis and SIRS datasets are available at figshare: Gucyetmez, Bulent; K. Atalan, Hakan (2016). Dataset. PLOS ONE. Dataset. https://doi.org/10.1371/journal.pone.0148699.s001.

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
