# Peer review of "The DBCV index is more informative than DCSI, CDbw, and VIASCKDE indices for unsupervised clustering internal assessment of concave-shaped and density-based clusters"

_PeerJ Computer Science, doi:10.7717/peerj-cs.3095_

## Round 0.1 · original submission · Major Revisions

The reviewers have substantial concerns about this manuscript. The authors should provide point-to-point responses to address all the concerns and provide a revised manuscript with the revised parts being marked in different color.

Reviewer 1 ·

Basic reporting

This paper compares the DBCV index (Moulavi et al. 2014) against six other cluster validity indices to determine the best one for detecting concave-shaped and nested clusters using DBSCAN as the base clustering technique. Unfortunately, this comparative study lacks novelty and originality since it is well-known that density-based indices (like DBCV) outperform centroid-based indices (like Davis-Bouldin, Calinski-Harabaz, and Gap statistics) and inter-point-based indices (like Silhouette and Dunn) in arbitrary and irregular-shaped clusters. Density-based indices measure a connectivity criterion, usually based on minimum spanning trees (MST), to characterize the cluster intrinsic structure, so comparing the DBCV index against indices that only measure the intra-cluster cohesion and inter-cluster separability is unfair. Besides, a centroid-based index can only detect globular-shaped clusters (because of the isotropic behavior of the Euclidean distance); thus, they are useful in techniques like k-means but not in density-based clustering algorithms.

Experimental design

- Provide the reasoning behind selecting the six indices compared against the DBCV index.

- All the evaluated indices should be mathematically described in addition to the DBCV index. Also, indicate their operative ranges and if they are maximized or minimized.

Validity of the findings

- The DBCV index should be compared and validated against density-based indices, preferentially against more recent approaches such as 10.1016/j.asoc.2020.106583, 10.1109/TNNLS.2018.2853710, and 10.1109/ACCESS. 2019.2906949.

- The results showing that DBCV outperformed non-connectivity-based indices are not meaningful; this is expected to happen because of their different natures.

Additional comments

- The title should be more concise.

- Some references are missing or incorrectly compiled (symbol "?" ).

Reviewer 2 ·

Basic reporting

-

Experimental design

-

Validity of the findings

-

Additional comments

1. The title is excessively long, overly specific, and borderline biased. Scientific titles should be concise, neutral, and informative, allowing readers to understand the topic without assuming the conclusion. This title asserts superiority of the DBCV index over all other indices listed, without acknowledging the experimental context or limitations.

Suggested Revision:
“Evaluating the Effectiveness of the DBCV Index for Unsupervised Clustering of Concave-Shaped and Nested Clusters”
or
“Comparative Assessment of Internal Clustering Indices for Complex Cluster Shapes: Focus on DBCV”

2. While the paper does not propose a new index, it focuses on empirically benchmarking the DBCV index against several established internal validation metrics in challenging clustering scenarios (e.g., concave and nested shapes). This is useful but does not constitute methodological novelty. But since this journal doesn't consider novelty, then it should be alright.

3. Methodological Concerns
3.1. Bias in framing
Throughout the manuscript, there is a strong bias toward promoting DBCV. The analysis often implies that DBCV is objectively better without adequately acknowledging:
+ The diversity of clustering contexts where other indices may perform better.
+ The design assumptions behind DBCV align well with density-based methods like DBSCAN.

3.2. Experimental design
+ The paper uses synthetic datasets with specific structures (concave, nested, manifold-based), which favor density-aware methods like DBCV.
+ There is no control for shape-neutral or centroid-based clusters (e.g., spherical, isotropic), where other indices might outperform DBCV.
I think the authors should include a broader dataset spectrum or clarify that the conclusion only applies to a specific type of cluster geometry.

3.3. Index implementation & metrics
+ It's unclear how consistently each index is applied across methods. Some (e.g., Gap Statistic, Shannon Entropy) are sensitive to parameters and assumptions.
+ Are all indices calculated based on the same clustering algorithm? For fair comparison, the same clustering outputs must be used.
+ The paper should report runtime and scalability differences as well.

4. Interpretation of results
+ The claim that DBCV is “more informative” is qualitative and not well defined. Does "informative" mean more correlated with ground-truth ARI/NMI? More stable across runs? Better for visualization?
+ While ARI and NMI are used for evaluation, they are external indices, which weakens the argument that DBCV is superior as an internal index.
You should consider using correlation analysis between internal and external indices (e.g., Spearman rank) to justify claims of alignment with clustering quality.

5. Other problems
+ Many sentences are verbose, redundant, or grammatically awkward. Please revise for conciseness and avoid subjective qualifiers.
+ Figures lack detailed captions and legends explaining the experiment context, data shape, and interpretation.
+ Some plots are cluttered or poorly labeled (e.g., axes with vague terms).
+ Provide pseudo-code or a structured explanation for how DBCV is computed.
+ Clarify if results generalize beyond DBSCAN or density-based clustering algorithms.

---

## Round 0.2 · accepted · Accept

Reviewers are satisfied with the revisions, and I concur in recommending accepting this manuscript.

Reviewer 2 ·

Basic reporting

N/A

Experimental design

N/A

Validity of the findings

N/A

Additional comments

Thank you to the authors for revising and updating the manuscript based on my questions and suggestions. This is a significant effort, as the authors had to redo a large amount of work in a short period of time. Although I believe the title of the paper is still not very strong, I think this ultimately depends on the authors' preference. As for the content, I consider this a fairly good paper in its revised version.